# *Withania somnifera* and *Chrysanthemum zawadskii* Herbich var. *latilobum* (Maxim.) Kitamura Complex Attenuates Obesity in High-Fat-Diet-Induced Obese Mice

**DOI:** 10.3390/ijms26115230

**Published:** 2025-05-29

**Authors:** Seong-Hoo Park, Jeongjin Park, Eunhee Yoo, Jaeeun Jung, Mi-Ryeong Park, Soyoung Kim, Jong-Lae Kim, Jong Wook Lee, Ok-kyung Kim, Minhee Lee

**Affiliations:** 1Clinical Nutrition Institute, Kyung Hee University, Seoul 02447, Republic of Korea; phoo3166@khu.ac.kr (S.-H.P.); a01089219455@gmail.com (E.Y.); jaeeun88@khu.ac.kr (J.J.); 2Division of Food and Nutrition and Human Ecology Research Institute, Chonnam National University, Gwangju 61186, Republic of Korea; pjj8425@hanmail.net; 3HLscience Co., Ltd., Uiwang-si 16004, Republic of Korea; alfud1030@hlscience.com (M.-R.P.); sykim@hlscience.com (S.K.); kimjl@hlscience.com (J.-L.K.); jwlee@hlscience.com (J.W.L.); 4Department of Food Innovation and Health, Kyung Hee University, Yongin 17104, Republic of Korea

**Keywords:** obesity, Ashwagandha (*Withania somnifera*), Gujeolcho, *Chrysanthemum zawadskii* Herbich var. *latilobum* (Maxim.) Kitamura

## Abstract

This study aims to evaluate the anti-obesity effects of Ashwagandha (*Withania somnifera*, AS), *Chrysanthemum zawadskii* Herbich var. *latilobum* (Maxim.) Kitamura (C), and their combination (AS:C = 3:1, ASC) in high-fat-diet (HFD)-induced obese animal models. Key metabolic parameters, including body weight, lipid metabolism, adipogenesis, energy expenditure, and glucose homeostasis, were assessed. HFD-fed mice were supplemented with AS25, C25, or ASC at different concentrations (ASC25, ASC50, and ASC100). Body weight, food efficiency ratio (FER), organ and adipose tissue weights were measured. Serum biochemical markers, including lipid profiles, glucose, insulin, and liver enzymes, were analyzed. Western blot analysis was conducted to assess the expression of key proteins involved in adipogenesis, lipogenesis, lipolysis, and energy metabolism. ASC complex supplementation, particularly at higher doses (ASC100), significantly reduced body weight gain, liver weight, and total white adipose tissue (WAT) accumulation. ASC complex groups exhibited improved lipid profiles, with reductions in triglycerides, total cholesterol, and low-density lipoprotein (LDL). Serum glucose, insulin, and HbA1c levels were significantly reduced, suggesting improved insulin sensitivity. Western blot analysis revealed that ASC complex supplementation downregulated key adipogenic markers, including PPARγ, C/EBPα, and SREBP1c, while enhancing adiponectin levels. ASC complex also promoted energy metabolism by increasing the phosphorylation of AMPK and UCP1 expression, indicative of enhanced thermogenesis and lipid oxidation. ASC complex supplementation demonstrates a potent anti-obesity effect by modulating adipogenesis, lipid metabolism, and energy expenditure. The findings suggest that ASC complex could serve as a promising natural therapeutic strategy for obesity and metabolic disorders. Further research, including clinical trials, is warranted to validate its efficacy and safety in human populations.

## 1. Introduction

Obesity has emerged as a major global health concern, with its prevalence rising at an alarming rate worldwide. It is a complex metabolic disorder associated with a range of comorbidities, including cardiovascular diseases, type 2 diabetes, non-alcoholic fatty liver disease (NAFLD), and certain types of cancer. The increasing incidence of obesity is largely attributed to excessive calorie intake, sedentary lifestyles, and genetic predisposition. Understanding the mechanisms underlying obesity and exploring novel therapeutic interventions remain key areas of scientific research [1,2,3]. To study obesity and its potential interventions, researchers commonly employ high-fat-diet (HFD)-induced animal models, which closely mimic the dietary patterns contributing to human obesity. These models typically involve feeding rodents diets containing 45–60% of calories from fat, significantly higher than the 10% fat content found in normal rodent diets, leading to increased body weight, adipose tissue accumulation, and metabolic disturbances similar to those observed in human obesity [4,5].

In recent years, there has been growing interest in natural compounds for their potential anti-obesity effects, particularly those derived from traditional herbal medicine. Such alternatives are gaining attention due to their efficacy, minimal side effects, and multi-targeted approach [6]. Among various medicinal plants, Ashwagandha (*Withania somnifera*) and *Chrysanthemum zawadskii* Herbich var. *latilobum* (Maxim.) Kitamura (commonly known as Gujeolcho in Korea) have been studied for their diverse pharmacological effects. Ashwagandha is a well-known adaptogenic herb traditionally used in Ayurvedic medicine to enhance physical endurance, reduce stress, and improve metabolic health [7]. It possesses anti-inflammatory, antioxidant, and metabolic regulatory properties, which may contribute to weight management. Previous studies have shown that Ashwagandha exerts anti-inflammatory effects via suppression of the NF-κB and MAPK pathways, and antioxidant effects via activation of the Nrf2 signaling pathway, which are relevant to metabolic homeostasis. Studies have shown that Ashwagandha can modulate cortisol levels, reduce fat accumulation, and improve insulin sensitivity, making it a promising candidate for obesity management [8,9,10,11]. *Chrysanthemum zawadskii* Herbich var. *latilobum* (Maxim.) Kitamura, a medicinal plant widely used in East Asian traditional medicine, has been reported to exhibit anti-obesity, anti-inflammatory, and lipid-lowering effects. It contains bioactive compounds such as flavonoids and phenolic acids, known to regulate lipid metabolism and reduce oxidative stress. Additionally, *Chrysanthemum zawadskii* Herbich var. *latilobum* (Maxim.) Kitamura has been used to treat respiratory ailments, inflammatory conditions, and gastrointestinal disorders, further highlighting its broad medicinal potential [12,13]. Despite the individual health benefits of Ashwagandha and *Chrysanthemum zawadskii* Herbich var. *latilobum* (Maxim.) Kitamura, their combined effects on obesity and metabolic disorders have not been extensively explored. Given their complementary pharmacological properties, investigating their synergistic effects could provide valuable insights into natural obesity management strategies.

The present study aims to evaluate the potential therapeutic effects of a combination of Ashwagandha (AS) and *Chrysanthemum zawadskii* Herbich var. *latilobum* (Maxim.) Kitamura (C) on HFD-induced obesity in an animal model. By assessing body weight, lipid metabolism, inflammatory markers, and other metabolic parameters, this study seeks to elucidate the mechanisms by which these herbal extracts influence metabolic health.

## 2. Results

### 2.1. Effects of ASC on Body Weight, Food Intake, FER, and Organ and Adipose Tissue Weight in HFD-Induced Obese Mice

To investigate the effects of AS25, C25, and ASC complex supplementation on obesity-related parameters, including body weight, food efficiency ratio (FER), organ weights, and adipose tissue weights were measured in HFD-fed animal models. The results are summarized in Table 1.

The initial body weights were not significantly different across all groups, indicating comparable baseline conditions before dietary intervention. However, after the experimental period, the final body weight was significantly increased in the CON (HFD control) group compared to the NC (normal control) group (*p* < 0.05). Among the treatment groups, ASC100 exhibited the lowest final body weight, significantly lower than the CON, AS25, and C25 groups, suggesting that higher ASC complex (ASC25, ASC50, and ASC100) supplementation attenuates obesity-related weight gain (8.9% min to 19.8% max) (*p* < 0.05). The weight gain followed a similar trend, with the CON group showing the highest weight gain, while the NC group exhibited the lowest. The treatment groups showed a dose-dependent reduction in weight gain, with ASC100 demonstrating the most significant decrease among the ASC complex groups (14.5% min to 33.5% max) (*p* < 0.05). FER, which represents the efficiency of food intake in promoting weight gain, was significantly higher in the CON group compared to the NC group (*p* < 0.05).

The CON group exhibited a significantly higher liver weight compared to the NC group, indicating liver hypertrophy due to HFD-induced obesity. The treatment groups showed a partial reduction in liver weight, with ASC100 displaying the most significant decrease (*p* < 0.05). No significant differences were observed in kidney and spleen weights across all groups, suggesting that HFD and supplementation did not significantly affect these organs. To assess adipose accumulation, total white adipose tissue (WAT), subcutaneous WAT, and visceral WAT were measured. The CON group showed a significant increase in total WAT weight compared to the NC group (*p* < 0.05). The treatment groups (AS25, C25, and ASC complex) exhibited reduced WAT weight, with ASC100 demonstrating the most significant reduction, indicating its potential to suppress adipogenesis (AS25; 21.2%, C25; 17.1%, ACS complex; 26.1% min to 47.6% max). A similar trend was observed, with the ASC100 group showing a significant reduction in subcutaneous fat compared to CON and other treatment groups (ASC complex; 25.1% min to 40.3% max) (*p* < 0.05). Visceral fat accumulation was highest in the CON group, while the ASC100 group exhibited the lowest visceral fat weight, suggesting an anti-obesity effect (ASC complex; 20.5% min to 44.2% max) (*p* < 0.05). These results suggest that the ASC complex supplementation, especially at the ASC100 dose, effectively alleviated HFD-induced obesity by reducing liver hypertrophy, visceral and subcutaneous fat accumulation, and overall adiposity, while demonstrating a synergistic anti-obesity effect of AS and C.

### 2.2. Effects of ASC on Biochemical Parameters in HFD-Induced Obese Mice

To investigate the effects of AS25, C25, and ASC complex (ASC25, ASC50, and ASC100) supplementation in high-fat diet (HFD)-induced obese animal models, key biochemical markers were analyzed in both serum and feces. The results are summarized in Table 2. Liver enzyme levels, alanine aminotransferase (ALT) and aspartate aminotransferase (AST), were significantly elevated in the CON group compared to the NC group (*p* < 0.05), indicating hepatic stress due to obesity. The ASC complex groups (ASC25, ASC50, and ASC100) exhibited a significant reduction in ALT and AST levels (ALT; 30.3% min to 38.0% max, AST; 19.8% min to 21.2% max), with ASC100 showing the greatest improvement, suggesting hepatoprotective effects. Triglyceride levels were markedly increased in the CON group compared to the NC group (*p* < 0.05), reflecting dyslipidemia induced by HFD. Treatment with AS25, C25, and ASC complex (ASC25, ASC50, and ASC100) significantly reduced serum triglyceride levels (AS25; 35.4%, C25; 20.8%, ASC complex; 36.3% min to 54.9% max), with ASC100 demonstrating the lowest triglyceride levels, indicating improved lipid metabolism. Total cholesterol levels were highest in the CON group and significantly reduced in the ASC groups, with ASC100 showing the greatest reduction (20.6% min to 38.2% max). Low-density lipoprotein (LDL-chol) was significantly higher in the CON group, indicating an increased risk of cardiovascular complications. Treatment with AS25, C25, and ASC (ASC25, ASC50, and ASC100) significantly reduced LDL levels, with ASC100 showing the most pronounced reduction (24.6% min to 45.8% max) (*p* < 0.05). High-density lipoprotein (HDL-chol) levels were slightly increased in the CON group compared to NC. However, the LDL/HDL ratio was significantly increased in the CON group (*p* < 0.05), whereas ASC complex supplementation reduced this ratio (19.0% min to 28.6% max), suggesting a protective effect against dyslipidemia. The CON group exhibited significantly elevated free fatty acid (FFA) levels. Treatment with AS25 and ASC significantly reduced FFA levels (AS25; 24.8%, ASC complex; 25.1% min to 31.3% max), with ASC100 showing the lowest levels, suggesting improved fatty acid metabolism. Serum glucose levels were significantly elevated in the CON group, indicative of insulin resistance. The ASC complex groups exhibited significant reductions in glucose levels, with ASC100 showing the greatest improvement (16.3% min to 30.2% max), suggesting enhanced glucose homeostasis. Serum insulin levels were markedly increased in the CON group, further indicating insulin resistance. Treatment with AS25, C25, and ASC complex significantly lowered insulin levels, with ASC100 showing a pronounced reduction (18.3% min to 44.5% max) (*p* < 0.05). HbA1c levels, an indicator of long-term glucose regulation, were significantly elevated in the CON group, while ASC complex supplementation significantly reduced HbA1c levels, with ASC100 demonstrating the lowest values (24.8% min to 36.9% max) (*p* < 0.05), suggesting improved glycemic control.

Fecal triglyceride levels were significantly higher in the ASC complex groups compared to the CON group (73.6% min to 196.3% max) (*p* < 0.05). ASC100 exhibited the highest fecal triglyceride levels, suggesting that ASC complex supplementation enhanced lipid excretion and reduced lipid absorption, contributing to improved lipid metabolism. Fecal cholesterol levels followed a similar pattern, with ASC complex groups, particularly ASC100, showing significantly increased cholesterol excretion compared to the CON group (78.3% min to 188.6% max) (*p* < 0.05). This finding suggests that ASC complex supplementation may enhance cholesterol clearance and prevent excessive lipid accumulation in the body.

### 2.3. Effects of AS, C, and ASC on Adipose and Muscle Mass in HFD-Induced Obese Mice

MicroCT analysis was performed to assess the impact of AS25, C25, and ASC complex (ASC25, ASC50, and ASC100) on fat and muscle mass in obese animal models fed a high-fat diet (HFD). The results demonstrate significant differences among the treatment groups (Figure 1). The ASC complex groups (ASC25, ASC50, and ASC100) exhibited a notable reduction in fat mass compared to the CON groups (21.1% min to 47.3% max). Additionally, the ASC50 and ASC100 resulted in an increase in muscle/adipose mass ratio compared to the CON groups (4.1% min to 34.4% max), further supporting their potential in modulating body composition in obesity models. In contrast, the AS25 and C25 single compounds showed a moderate reduction in fat mass, but their effects were less pronounced compared to the ASC complex (ASC25, ASC50, and ASC100). These findings indicate that the ASC complex (ASC25, ASC50, and ASC100) has a superior role in improving body composition in obese animal models.

### 2.4. Effects of AS, C, and ASC on Adipogenesis and Lipogenesis Pathways in HFD-Induced Obese Mice

To investigate the effects of AS25, C25, and ASC complex (ASC25, ASC50, and ASC100) on adipogenesis-related pathways in obese animal models fed a high-fat diet (HFD), Western blot analysis was performed to examine the expression levels of key adipogenic proteins, including p-MAPK, PPARγ, p-CREB/CREB, C/EBPα, SREBP1c, Leptin, and Adiponectin (Figure 2A). The p-MAPK was significantly decreased in the ASC complex groups compared to the CON group (36.4% min to 57.7% max) (*p* < 0.05). PPARγ, a key regulator of adipogenesis, was significantly increased in the CON groups compared to the NC (normal control) group (*p* < 0.05). The ASC complex groups (ASC25, ASC50, and ASC100) exhibited a dose-dependent decrease in PPARγ expression, with ASC100 showing the most pronounced reduction (6.3% min to 37.3%) (*p* < 0.05), suggesting its potential inhibitory effects on adipogenesis. The expression of p-CREB/CREB, a transcription factor involved in adipocyte differentiation, was significantly elevated in the CON group compared to the NC (*p* < 0.05). The expression levels of p-CREB were reduced in all treatment groups, with ASC100 exhibiting the lowest level among the ASC complex groups (ASC25, ASC50, and ASC100) (43.1% min to 65.2% max) (*p* < 0.05). The total CREB levels remained relatively constant across all groups, indicating that changes were specific to the phosphorylated form. C/EBPα, another crucial transcription factor for adipogenesis, was highly expressed in the CON group (*p* < 0.05). ASC complex groups (ASC25, ASC50, and ASC100) significantly reduced C/EBPα expression compared to the CON group, with ASC100 showing the most notable decrease (22.3% min to 43.0% max) (*p* < 0.05), further supporting its anti-adipogenic effects. SREBP1c expression followed a similar trend to other adipogenic markers, with the highest levels observed in the CON group (*p* < 0.05). The ASC complex groups (ASC25, ASC50, and ASC100) exhibited a significant decrease in SREBP1c expression (37.6% min to 55.3% max) (*p* < 0.05), suggesting a suppression of lipid synthesis and adipogenesis. Leptin, a hormone associated with adipocyte differentiation and lipid metabolism, was significantly elevated in the CON group compared to NC (*p* < 0.05). However, treatment with ASC complex resulted in a marked reduction in leptin levels, with ASC100 demonstrating the lowest expression (32.9% min to 51.3%) (*p* < 0.05). Conversely, adiponectin, an anti-inflammatory adipokine, was significantly reduced in the HFD-fed groups (*p* < 0.05). ASC complex groups (ASC25, ASC50, and ASC100) showed a significant increase in adiponectin expression (68.0% min to 188.2% max) compared to the CON group (*p* < 0.05), indicating its potential beneficial effects on adipocyte function.

To further examine the effects of AS25, C25, and ASC complex (ASC25, ASC50, and ASC100) on lipid metabolism and adipogenesis in obese animal models fed an HFD, we analyzed the expression of key lipogenic proteins, including glucose-6-phosphate dehydrogenase (G6PDH), fatty acid synthase (FAS), p-Acetyl-CoA carboxylase (ACC)/ACC, p-ATP-citrate lyase (ACL)/ACL, and lipoprotein lipase (LPL) using Western blot (Figure 2B). The expression of G6PDH, a key enzyme in the pentose phosphate pathway that supports lipogenesis, was significantly elevated in the CON group compared to the NC group (*p* < 0.05). However, treatment with AS25, C25, and ASC complex led to a significant reduction in G6PDH expression, with ASC100 showing the lowest expression levels (25.6% min to 51.8% max) (*p* < 0.05), suggesting that ASC supplementation may inhibit lipid biosynthesis. FAS, a critical enzyme in de novo lipogenesis, was highly expressed in the CON group, whereas the NC group showed the lowest expression (*p* < 0.05). However, treatment with AS25, C25, and ASC complex significantly reduced FAS expression, with ASC100 demonstrating the most pronounced suppression (AS25, 39.9%; C25, 44.1%; ASC, 39.2% min to 56.7% max) (*p* < 0.05), indicating a dose-dependent inhibition of FAS. ACC is a rate-limiting enzyme in fatty acid biosynthesis, and its phosphorylated form (p-ACC) is indicative of its inactivation. The ratio of p-ACC/ACC was significantly decreased in the CON group compared to the NC group (*p* < 0.05). Treatment with ASC complex (ASC25, ASC50, and ASC100) showed a significant increase in ACC phosphorylation (153.8% min to 406.5% max) (*p* < 0.05). Notably, ASC100 treatment significantly increased the p-ACC/ACC ratio, suggesting an inhibitory effect on fatty acid synthesis. ACL catalyzes the conversion of citrate to acetyl-CoA, a key precursor for fatty acid synthesis. The phosphorylated form (p-ACL) is an active state of the enzyme. The NC group exhibited the highest p-ACL/ACL ratio, whereas the CON showed significantly lower phosphorylation levels (*p* < 0.05, Figure 2B) indicating enhanced lipid biosynthesis in these groups. Treatment with AS25, C25, and ASC complex (ASC25, ASC50, and ASC100) significantly increased p-ACL/ACL levels, with ASC100 showing a notable increase (40.1% min to 610.3% max), suggesting that ASC complex may regulate lipid metabolism through ACL modulation. LPL plays a crucial role in lipid metabolism by hydrolyzing triglycerides for uptake by adipose tissue. The expression of LPL was significantly increased in the CON group compared to NC (*p* < 0.05). Treatment with AS25, C25, and ASC complex (ASC25, ASC50, and ASC100) significantly reduced LPL expression, with ASC100 showing the lowest levels (22.0% min to 64.9% max) (*p* < 0.05), suggesting a reduction in lipid uptake and storage in adipose tissue.

### 2.5. Effects of AS, C, and ASC on Lipolysis Pathway and Energy Metabolism in HFD-Induced Obese Mice

To evaluate the effects of AS single compound (AS), C single compound (C), and ASC complex (ASC) on lipolysis pathways in obese animal models fed with a high-fat diet (HFD), Western blot analysis was performed to measure the expression levels of key proteins involved in lipolysis, including cyclic adenosine monophosphate (cAMP), adipose triglyceride lipase (ATGL), protein kinase A (PKA), hormone-sensitive lipase (HSL), perilipin (PLIN), and phosphodiesterase 3B (PDE3B) (Figure 3A). cAMP levels were significantly increased treatment groups compared to the CON group (76.9% min to 209.9% max) (*p* < 0.05). Western blot analysis revealed a significant reduction in ATGL expression in the CON group compared to the NC group. Among the treatment groups, ASC50 and ASC100 exhibited significantly increased ATGL levels compared to the CON group (205.5% and 202.1%) (*p* < 0.05), with values comparable to the NC group. PKA expression was significantly reduced in the CON group compared to the NC group. Among the treatment groups, ASC50 and ASC100 significantly restored PKA levels (111.5% and 141.1%) (*p* < 0.05). The expression of HSL was also significantly downregulated in the CON group compared to the NC group. Treatment with ASC complex (ASC25, ASC50 and ASC100) significantly increased HSL expression to levels comparable to the CON group (126.6% min to 264.0% max) (*p* < 0.05). PLIN expression was significantly elevated in the CON group compared to the NC group, while the treatment with ASC complex (ASC25, ASC50 and ASC100) showed significant reductions in PLIN expression (13.9% min to 47.6% max) (*p* < 0.05). PDE3B expression was significantly elevated in the CON group compared to the NC group, while the treatment with AS25, C25, and ASC complex (ASC25, ASC50 and ASC100) showed significant reductions compared to the CON group (AS25; 40.8%, C25; 19.5%, ASC complex; 30.4% min to 49.3% max) (*p* < 0.05).

To assess the impact of AS25, C25, and ASC complex (ASC25, ASC50, and ASC100) on energy metabolism in obese animal models fed an HFD, we conducted Western blot analysis to examine the expression of key metabolic regulators, including p-AMP-activated protein kinase (AMPK)/AMPK, uncoupling protein 1 (UCP1), Carnitine palmitoyltransferase 1A (CPT1A), and fatty acid-binding protein 4 (FABP4) (Figure 3B). AMPK is a crucial regulator of cellular energy homeostasis. The ratio of p-AMPK/AMPK was significantly reduced in the HFD control (CON) group compared to the NC group, indicating impaired energy metabolism in obesity (*p* < 0.05, Figure 3B). Treatment with ASC complex (ASC25, ASC50, and ASC100) restored AMPK phosphorylation in a dose-dependent manner, with ASC100 showing the highest p-AMPK/AMPK ratio among treated groups (77.8% min to 370.1% max) (*p* < 0.05), suggesting that ASC enhances energy metabolism by activating AMPK signaling. UCP1 is a mitochondrial protein involved in thermogenesis and energy expenditure. UCP1 expression was significantly reduced in the CON compared to the NC group (*p* < 0.05), indicating a reduction in thermogenic capacity due to HFD-induced obesity. Treatment with ASC complex (ASC25, ASC50, and ASC100) partially restored UCP1 expression, with ASC100 showing a significant increase in UCP1 levels compared to lower doses (93.9% min to 257.9% max) (*p* < 0.05). These findings suggest that ASC may enhance energy expenditure and mitigate obesity-induced metabolic dysfunction. CPT1A is a key enzyme in mitochondrial fatty acid oxidation. The Western blot results indicate increased CPT1A expression in ASC complex groups compared to the CON group (173.2% min to 344.5% max) (*p* < 0.05). The observed increase in CPT1A levels suggests that ASC complex may promote lipid oxidation and reduce fat accumulation in adipose tissue. FABP4 is involved in lipid transport and metabolism, and its elevated expression is often associated with adipogenesis and metabolic disorders. The CON group exhibited significantly higher FABP4 expression compared to the NC group (*p* < 0.05), consistent with increased lipid accumulation in obesity. ASC complex (ASC25, ASC50, and ASC100) significantly reduced FABP4 levels, with ASC100 showing the most pronounced reduction (19.3% min to 49.6% max) (*p* < 0.05), suggesting that ASC may help regulate lipid metabolism and prevent excessive fat accumulation.

## 3. Discussion

Obesity is a multifactorial metabolic disorder characterized by excessive adipose tissue accumulation and metabolic dysfunction, which is primarily driven by an imbalance between energy intake and expenditure [14]. The present study demonstrates that supplementation with a combination of Ashwagandha (AS) and *Chrysanthemum zawadskii* Herbich var. *latilobum* (Maxim.) Kitamura (C) (ASC) effectively mitigates high-fat-diet (HFD)-induced obesity in mice by modulating body weight, lipid metabolism, adipogenesis, lipolysis, and energy metabolism.

Body weight and weight gain trends observed in this study indicate that HFD consumption leads to significant weight gain and increased FER, a finding consistent with previous research [15]. In the treated-with-ACS-complex groups, particularly ASC100, there was no significant difference in FER; however, the reduction in body weight suggests that ASC complex supplementation may enhance energy expenditure and reduce energy storage efficiency. Organ weight analysis revealed that HFD consumption resulted in liver hypertrophy, as evidenced by the significantly increased liver weight in the CON group. This finding is agreement with previous studies showing that HFD-induced obesity is associated with hepatic lipid accumulation and enlargement [16,17]. The reduction in liver weight observed in ASC100 group suggests that ASC complex supplementation may help mitigate hepatic lipid accumulation, thereby reducing liver hypertrophy. However, kidney and spleen weights were not significantly affected across groups, indicating that ASC complex supplementation does not exert adverse effects on these organs. Adipose tissue analysis further supports the anti-obesity potential of ASC100. The significant reduction in total WAT, subcutaneous WAT, and visceral WAT in ASC100 group suggests that ASC complex supplementation suppresses adipogenesis. Prior research has shown that adipose tissue accumulation, particularly visceral fat, is strongly associated with metabolic disorders, including insulin resistance and inflammation [18]. The reduction in visceral fat observed in the ASC100 group suggests a potential protective effect against obesity-related metabolic dysfunction.

Obesity-induced hepatic stress is commonly associated with elevated liver enzyme levels, particularly ALT and AST, which are markers of liver damage. In this study, ALT and AST levels were significantly higher in the CON group, indicating hepatic stress likely due to lipid accumulation in the liver (steatosis) [19,20]. However, ASC complex supplementation, particularly ASC100, significantly reduced these enzyme levels, suggesting hepatoprotective effects. HFD consumption led to significant dyslipidemia in the CON group, as evidenced by increased triglycerides, total cholesterol, and LDL-cholesterol levels. Elevated LDL-cholesterol is a well-known risk factor for cardiovascular diseases, while a high LDL/HDL ratio indicates an increased risk of atherosclerosis and metabolic syndrome [21,22]. The ASC complex supplementation, particularly ASC100, exhibited the most pronounced reductions in triglycerides, total cholesterol, and LDL-cholesterol levels, suggesting an improvement in lipid metabolism. These results are consistent with previous studies that demonstrated the lipid-lowering effects of bioactive compounds in ASC supplementation, which may enhance hepatic lipid clearance and reduce lipid absorption in the intestine [23,24]. Furthermore, the decrease in the LDL/HDL ratio observed with ASC supplementation suggests a potential cardioprotective effect, as a lower LDL/HDL ratio is associated with improved lipid profiles and reduced cardiovascular risk, with HDL playing a crucial role in reverse cholesterol transport and anti-inflammatory responses. FFAs play a key role in metabolic dysfunction and insulin resistance. The significantly elevated FFA levels observed in the CON group indicate impaired fatty acid metabolism, a hallmark of metabolic disorders associated with obesity [25]. ASC supplementation significantly reduced FFA levels, with ASC100 showing the greatest reduction. Obesity-induced insulin resistance is a major contributor to metabolic syndrome and type 2 diabetes. The CON group exhibited significantly higher serum glucose and insulin levels, as well as elevated HbA1c, indicating impaired glucose metabolism [26,27]. ASC supplementation effectively reduced these markers, with ASC100 showing the most pronounced improvements. A notable finding of this study is the increased fecal triglyceride and cholesterol levels in ASC supplementation groups, particularly ASC100. Increased fecal lipid excretion has been observed with dietary fiber and bioactive compounds that inhibit lipid digestion and absorption [28].

Adipogenesis and lipogenesis are complex processes regulated by various transcription factors and metabolic enzymes. PPAR-γ and C/EBP play central roles in adipocyte differentiation by activating adipogenic gene expression, while CREB contributes to early adipogenesis through cAMP signaling. SREBP is a key transcription factor in lipogenesis, regulating lipid metabolism by inducing enzymes such as FAS, ACC, and ACL, which are involved in fatty acid synthesis. Additionally, G6PDH provides NADPH for lipogenesis, supporting de novo lipid synthesis. LPL facilitates triglyceride hydrolysis and uptake, further promoting lipid accumulation. Adipokines such as leptin and adiponectin play crucial roles in lipid homeostasis and insulin sensitivity. Moreover, MAPK signaling influences adipogenesis by modulating the activity of PPAR-γ and C/EBP, thereby regulating extracellular signals into adipogenic differentiation. In particular, ERK1/2 activation has been shown to play a crucial role in the early stages of adipogenic commitment and differentiation. These regulatory networks collectively orchestrate adipose tissue development and lipid metabolism, balancing energy storage and expenditure in response to metabolic demands [29,30,31,32,33]. This study demonstrates that ASC complex supplementation exerts anti-adipogenic and lipid-regulating effects in HFD-induced obese animal models by modulating key signaling pathways involved in adipocyte differentiation and lipid metabolism. Western blot analysis revealed that ASC complex supplementation, particularly at higher doses (ASC100), significantly downregulated the expression of key adipogenic transcription factors, including PPARγ, C/EBPα, and SREBP1c, suggesting a suppression of adipogenesis. The reduction in *p*-CREB/CREB and leptin levels further supports the inhibition of adipocyte differentiation and lipid accumulation. Conversely, adiponectin expression, which is typically reduced in obesity, was partially restored in ASC complex groups, indicating potential benefits in improving insulin sensitivity and anti-inflammatory responses. In addition to inhibiting adipogenesis, ASC complex supplementation effectively modulated lipid metabolism by downregulating key lipogenic enzymes. The expression of G6PDH, FAS, and LPL—enzymes involved in fatty acid and triglyceride synthesis—was significantly reduced in ASC complex groups, particularly ASC100, suggesting a suppression of lipid biosynthesis and uptake. Furthermore, ASC complex supplementation increased the phosphorylation of ACC and ACL, indicating reduced lipogenesis and enhanced lipid oxidation. These molecular changes collectively suggest that ASC complex supplementation reduces lipid accumulation in adipose tissue while promoting lipid metabolism.

Lipolysis and energy metabolism are tightly regulated by hormonal and enzymatic mechanisms that control lipid mobilization and utilization. Lipolysis, the breakdown of triglycerides into free fatty acids and glycerol, is primarily mediated by HSL and ATGL, which are activated in response to PKA signaling. PLIN plays a crucial role in regulating lipolysis by shielding lipid droplets from hydrolysis under basal conditions and facilitating enzyme access upon phosphorylation. PDE3B modulates this process by degrading cAMP, thereby reducing PKA activation and suppressing lipolysis in response to insulin signaling. Released fatty acids are transported by FABP4 and utilized for β-oxidation, a process regulated by CPT1A, which facilitates mitochondrial fatty acid uptake. In energy-demanding conditions, AMPK acts as a metabolic sensor, inhibiting lipogenesis and promoting fatty acid oxidation to sustain ATP production. In brown adipose tissue, UCP1 enhances energy dissipation through thermogenesis, linking lipolysis to adaptive energy expenditure. These regulatory pathways highlight the intricate balance between lipid mobilization, oxidation, and thermogenesis, ensuring metabolic homeostasis in response to energy demands [34,35,36]. This study demonstrates that ASC complex supplementation enhances lipolysis and energy metabolism in HFD-induced obese animal models by modulating key lipid breakdown and mitochondrial function pathways. Western blot analysis revealed that ASC complex, particularly at higher doses (ASC50 and ASC100), significantly increased the expression of ATGL, PKA, and HSL, promoting triglyceride hydrolysis, while reducing PLIN expression, thereby facilitating lipid mobilization. Additionally, ASC supplementation restored AMPK activation, with ASC100 showing the highest *p*-AMPK/AMPK ratio, indicating improved energy metabolism and fatty acid oxidation. The upregulation of UCP1 and CPT1A in ASC complex supplementation suggests enhanced thermogenesis and mitochondrial lipid utilization, contributing to increased energy expenditure and reduced adiposity. Moreover, the significant reduction in FABP4 expression in ASC complex supplementation suggests that ASC may help regulate lipid transport and prevent excessive fat accumulation. Collectively, these findings indicate that ASC complex supplementation exerts anti-obesity effects through a multi-faceted mechanism involving enhanced lipolysis, increased fatty acid oxidation, and improved mitochondrial function, with ASC100 showing the most pronounced effects. These results highlight the potential of ASC complex as a functional intervention for obesity management, warranting further studies to explore its clinical applicability.

## 4. Materials and Methods

### 4.1. Preparation of Withania somnifera and Chrysanthemum zawadskii Herbich Var. Latilobum (Maxim.) Kitamura Complex (ASC) and Quantitative of Withanoside IV and Linarin

The ASC was provided by HL science Co., Ltd. (Uiwang, Republic of Korea). The AS was prepared by extracting Ashwagandha (*Withania somnifera*) roots with 80% ethanol, followed by concentration and drying to obtain Ashwagandha extract powder. The C was prepared by extracting the whole plant of *Chrysanthemum zawadskii* Herbich var. *latilobum* (Maxim.) Kitamura with 50% ethanol, followed by concentration and drying to obtain *Chrysanthemum zawadskii* Herbich var. *latilobum* (Maxim.) Kitamura extract powder. ASC was formulated by combining Ashwagandha extract and *Chrysanthemum zawadskii* Herbich var. *latilobum* (Maxim.) Kitamura extract in a 3:1 ratio. The HPLC system consisted of Waters e2695 system equipped with a 2998 PDA detector (Waters Corporation, Milford, MA, USA), and Empower 3 software (Waters Corporation) was used. Withanoside IV was performed using Capcellpak C18 MG column, 4.6 × 250 mm, 5 μm. The analyte was eluted 0.1% phosphoric acid in water (*v*/*v*) and acetonitrile with stepwise elution mode. Column temperature was set at 27 °C and the flow rate was 1.0 mL/min. The detection wavelength was set at 227 nm. Linarin was performed using Zorbax Eclipse plus C18 column, 4.6 × 250 mm, 5 μm. The analyte was eluted with 0.1% formic acid in water (*v*/*v*) and 0.1% formic acid in acetonitrile with stepwise elution mode. Column temperature was set at 30 °C and the flow rate was 1.0 mL/min. The detection wavelength was set at 310 nm. The content of standardized compounds in ASC was determined to be Withanoside IV 4.60 mg/g ± 20% and Linarin 5.11 mg/g ± 20% (Figure 4).

### 4.2. Animal

C57BL/6J mice (4-week-old mice, male) were purchased from Saeron Bio (Uiwang, Republic of Korea) and housed in cages under controlled conditions (22 ± 2 °C, 55% humidity, and a 12:12 h light–dark cycle). Mice were allowed to adapt to conditions for one week. Thereafter, they were fed with an NC (normal control, AIN93G normal diet), CON (60% high-fat diet, HFD), MF (HFD containing metformin 250 mg/kg b.w.), AS25 (HFD containing *Withania somnifera* 25 mg/kg b.w.), C25 (HFD containing *Chrysanthemum zawadskii* Herbich var. *latilobum* (Maxim.) Kitamura 25 mg/kg b.w.), ASC complex (HFD containing combination of *Withania somnifera* and *Chrysanthemum zawadskii* Herbich var. *latilobum* (Maxim.) Kitamura [3:1] 25, 50, and 100 mg/kg b.w.) for 14 weeks. Mice were killed and their tissues and blood (by orbital venipuncture) were collected. The experimental protocol was approved by the Institutional Animal Care and Use Committee of Kyung Hee University (KHGASP-22-571).

### 4.3. Biochemical Factors in Serum and Feces

The levels of ALT, AST, triglycerides, total cholesterol, LDL-cholesterol, HDL-cholesterol, FFA, glucose (all from cam, Cambridege, UK), insulin (ALPCO, Salem, NH, USA), and HbA1c (Crystal Chem, Elk Grove Village, IL, USA) were measured in serum and feces using ELISA Kit according to the respective assay kit manufacturers’ instructions.

### 4.4. Micro-CT

Mice were examined by whole via abdominal tomography using micro-CT equipment (VIVA CT 80, Scano Medical AG, Brüttisellen, Switzerland).

### 4.5. Western Blot

Proteins were extracted from the epididymal adipose tissues and brown adipose tissues of mice were analyzed for the expressions of p-MAPK (Erk1/2), *p*-CREB, CREB, SREBP1c, PPAR-γ, C/EBPα, Leptin, adiponectin, G6PDH, *p*-ACL, ACL, *p*-ACC, ACC, FAS, LPL, PKA, PDE3B, HSL, PLIN, ATGL, *p*-AMPK, AMPK, UCP1, CPT1A, FABP4, and β-actin according to methods described previously [37].

### 4.6. Statistical Analysis

All data are expressed as the mean ± standard deviation (SD). Significant differences were determined using one-way analysis of variance (ANOVA) and Duncan’s multiple range test (SPSS PASW Statistic v.23.0, SPSS Inc., Chicago, IL, USA). Statistical significance was determined at *p* < 0.05.

## 5. Conclusions

In summary, our study provides compelling evidence that ASC supplementation exerts anti-obesity effects by reducing body weight gain, improving lipid metabolism, suppressing adipogenesis and lipogenesis, and enhancing lipolysis and energy metabolism. The combined effects of AS and C appear to produce a synergistic impact on obesity-related metabolic pathways. Given these promising findings, ASC supplementation may be a potential natural therapeutic strategy for preventing and managing obesity. Future studies should further investigate the molecular mechanisms underlying these effects, focusing on key biomarkers identified in this study, and evaluate the long-term safety and efficacy of ASC in clinical settings.

## Figures and Tables

**Figure 1 ijms-26-05230-f001:**
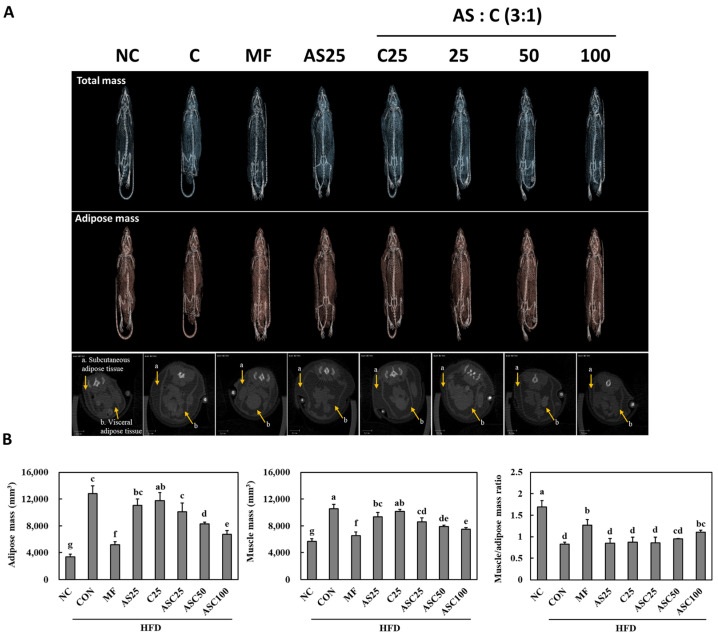
Effect of ASC complex on adipose and muscle mass of HFD-induced obese mice. MicroCT analysis (**A**) of adipose and muscle mass (**B**). NC, normal control; CON, obesity-induced control; MF, Metformin 250 mg/kg b.w., positive control; AS25, Ashwagandha (*Withania somnifera*) 25 mg/kg b.w., C25, *Chrysanthemum zawadskii* Herbich var. *latilobum* (Maxim.) Kitamura 25 mg/kg b.w.; ASC25, Ashwagandha (*Withania somnifera*): *Chrysanthemum zawadskii* Herbich var. *latilobum* (Maxim.) Kitamura (3:1) 25 mg/kg b.w.; ASC50, Ashwagandha (*Withania somnifera*): *Chrysanthemum zawadskii* Herbich var. *latilobum* (Maxim.) Kitamura (3:1) 50 mg/kg b.w.; ASC100, Ashwagandha (*Withania somnifera*): *Chrysanthemum zawadskii* Herbich var. *latilobum* (Maxim.) Kitamura (3:1) 100 mg/kg b.w. Scale bar: 5 mm. Values are presented as mean ± standard deviation (*n* = 8), and different superscript letters indicate significance at *p* < 0.05.

**Figure 2 ijms-26-05230-f002:**
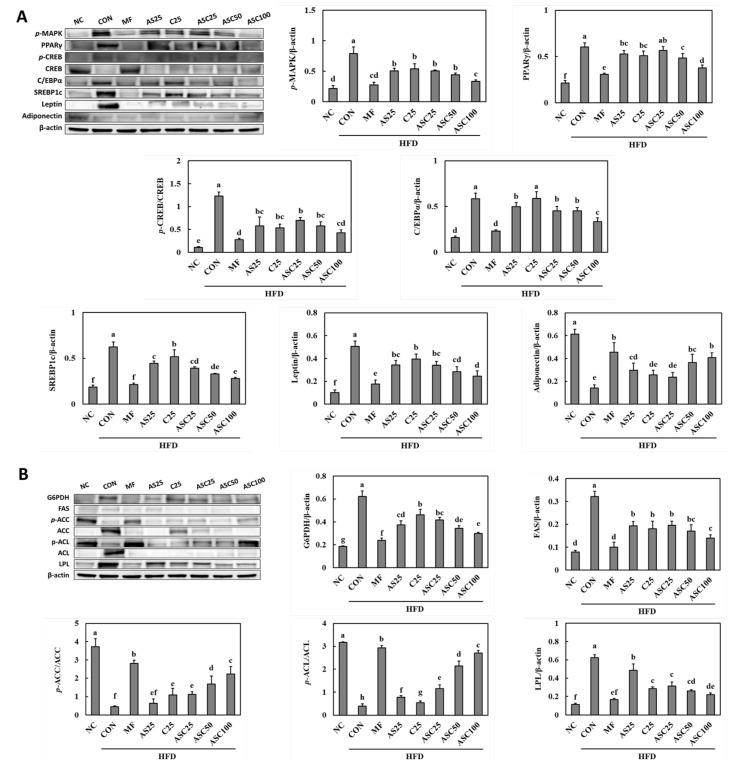
Effect of ASC complex on adipogenesis and lipogenesis pathways of HFD-induced obese mice. (**A**) Adipogenesis pathways including protein expression, (**B**) lipogenesis pathway including protein expression. NC, normal control; CON, obesity-induced control; MF, Metformin 250 mg/kg b.w., positive control; AS25, Ashwagandha (*Withania somnifera*) 25 mg/kg b.w., C25, *Chrysanthemum zawadskii* Herbich var. *latilobum* (Maxim.) Kitamura 25 mg/kg b.w.; ASC25, Ashwagandha (*Withania somnifera*): *Chrysanthemum zawadskii* Herbich var. *latilobum* (Maxim.) Kitamura (3:1) 25 mg/kg b.w.; ASC50, Ashwagandha (*Withania somnifera*): *Chrysanthemum zawadskii* Herbich var. *latilobum* (Maxim.) Kitamura (3:1) 50 mg/kg b.w.; ASC100, Ashwagandha (*Withania somnifera*): *Chrysanthemum zawadskii* Herbich var. *latilobum* (Maxim.) Kitamura (3:1) 100 mg/kg b.w. Values are presented as mean ± standard deviation (*n* = 8), and different superscript letters indicate significance at *p* < 0.05.

**Figure 3 ijms-26-05230-f003:**
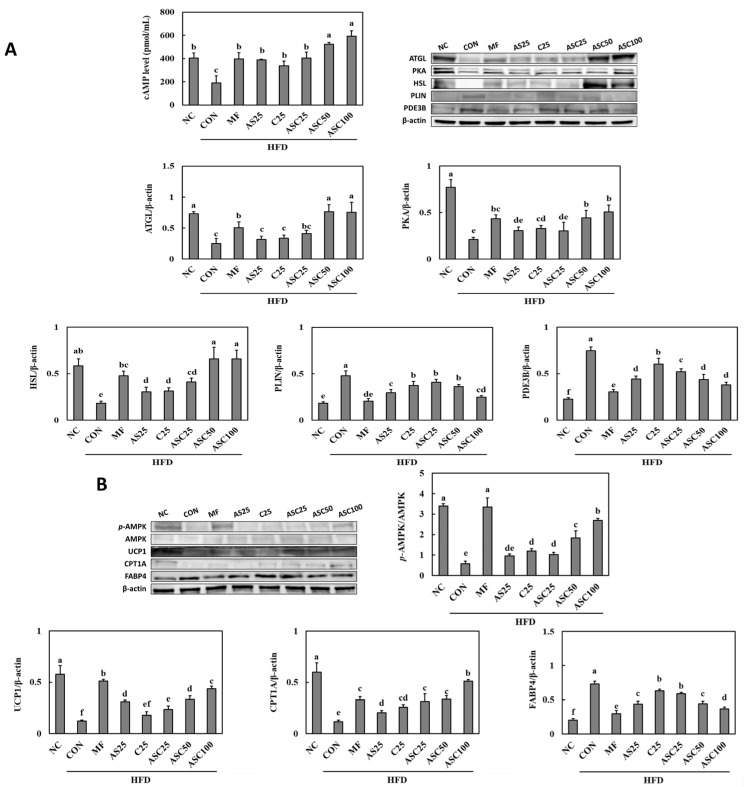
Effect of ASC complex on lipolysis pathways and energy metabolism of HFD-induced obese mice. (**A**) Lipolysis pathways including protein expression, (**B**) energy metabolism including protein expression. NC, normal control; CON, obesity-induced control; MF, Metformin 250 mg/kg b.w., positive control; AS25, Ashwagandha (*Withania somnifera*) 25 mg/kg b.w., C25, *Chrysanthemum zawadskii* Herbich var. *latilobum* (Maxim.) Kitamura 25 mg/kg b.w.; ASC25, Ashwagandha (*Withania somnifera*): *Chrysanthemum zawadskii* Herbich var. *latilobum* (Maxim.) Kitamura (3:1) 25 mg/kg b.w.; ASC50, Ashwagandha (*Withania somnifera*): *Chrysanthemum zawadskii* Herbich var. *latilobum* (Maxim.) Kitamura (3:1) 50 mg/kg b.w.; ASC100, Ashwagandha (*Withania somnifera*): *Chrysanthemum zawadskii* Herbich var. *latilobum* (Maxim.) Kitamura (3:1) 100 mg/kg b.w. Values are presented as mean ± standard deviation (*n* = 8), and different superscript letters indicate significance at *p* < 0.05.

**Figure 4 ijms-26-05230-f004:**
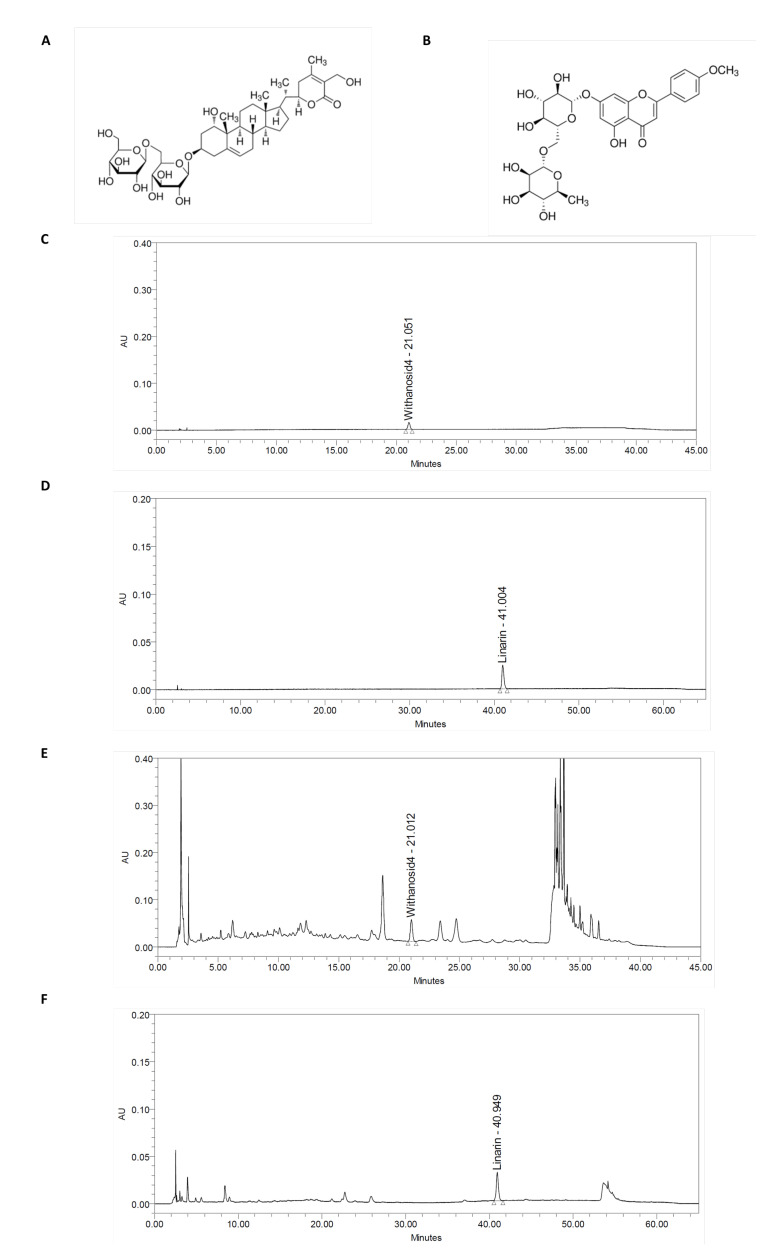
HPLC chromatograms of Withanoside IV and Linarin in ASC complex at 227 or 310 nm. (**A**) Withanoside IV and (**B**) Linarin structure, (**C**) Withanoside IV and (**D**) Linarin standard chromatogram, (**E**) ASC complex chromatogram for Withanoside IV and (**F**) Linarin.

**Table 1 ijms-26-05230-t001:** Effects of ASC complex on body weight, food intake, FER, organ and adipose tissue weight in HFD-induced obese mice.

	NC	CON	MF	AS25	C25	ASC25	ASC50	ASC100
Initial body weight (g)	20.97 ± 0.75 ^ns^	21.16 ± 1.09	20.59 ± 0.64	20.71 ± 0.92	21.03 ± 1.37	20.81 ± 0.87	20.81 ± 1.09	20.66 ± 0.84
Final body weight (g)	32.23 ± 1.78 ^f^	48.11 ± 2.86 ^a^	35.47 ± 0.79 ^e^	45.27 ± 2.76 ^ab^	46.19 ± 4.26 ^ab^	43.83 ± 3.21 ^bc^	40.83 ± 2.28 ^cd^	38.57 ± 2.38 ^d^
Weight gain * (g)	11.26 ± 1.51 ^f^	26.94 ± 2.71 ^a^	14.88 ± 1.33 ^e^	24.56 ± 2.58 ^ab^	25.16 ± 5.88 ^ab^	23.02 ± 3.48 ^bc^	20.02 ± 2.28 ^cd^	17.91 ± 2.48 ^de^
FER **	5.02 ± 0.87 ^b^	8.69 ± 0.87 ^a^	7.59 ± 0.68 ^a^	8.61 ± 0.90 ^a^	8.73 ± 1.80 ^a^	8.93 ± 1.35 ^a^	8.62 ± 0.98 ^a^	7.92 ± 1.09 ^a^
Organ weight (g)								
Liver	1.28 ± 0.10 ^f^	2.43 ± 0.16 ^a^	1.49 ± 0.12 ^e^	1.97 ± 0.24 ^bc^	2.13 ± 0.24 ^b^	1.82 ± 0.20 ^cd^	1.79 ± 0.15 ^de^	1.62 ± 0.15 ^de^
Kidney	0.40 ± 0.03 ^ns^	0.42 ± 0.05	0.41 ± 0.06	0.41 ± 0.04	0.42 ± 0.03	0.41 ± 0.02	0.42 ± 0.03	0.40 ± 0.01
Spleen	0.12 ± 0.03 ^ns^	0.13 ± 0.01	0.13 ± 0.04	0.11 ± 0.02	0.12 ± 0.03	0.11 ± 0.02	0.11 ± 0.03	0.11 ± 0.02
Adipose tissue weight (g)								
Total WAT ***	1.10 ± 0.19 ^g^	6.26 ± 0.39 ^a^	2.50 ± 0.29 ^f^	4.93 ± 0.34 ^bc^	5.15 ± 0.37 ^b^	4.62 ± 0.35 ^c^	3.95 ± 0.45 ^d^	3.28 ± 0.30 ^e^
Subcutaneous WAT	0.48 ± 0.11 ^f^	2.90 ± 0.20 ^a^	0.93 ± 0.24 ^e^	2.15 ± 0.19 ^b^	2.17 ± 0.29 ^b^	1.95 ± 0.21 ^bc^	1.73 ± 0.14 ^c^	1.40 ± 0.23 ^d^
Visceral WAT	0.63 ± 0.12 ^g^	3.36 ± 0.38 ^a^	1.57 ± 0.18 ^f^	2.78 ± 0.24 ^bc^	2.98 ± 0.23 ^b^	2.67 ± 0.25 ^c^	2.23 ± 0.36 ^d^	1.87 ± 0.22 ^e^

* Weight gain = final body weight − initial body weight; ** FER, food efficiency rate = weight gain (g)/total food consumption (g) × 100; *** WAT, white adipose tissue; ns, no significant difference. NC, normal control; CON, obesity-induced control; MF, Metformin 250 mg/kg b.w., positive control; AS25, Ashwagandha (*Withania somnifera*) 25 mg/kg b.w., C25, *Chrysanthemum zawadskii* Herbich var. *latilobum* (Maxim.) Kitamura 25 mg/kg b.w.; ASC25, Ashwagandha (*Withania somnifera*): *Chrysanthemum zawadskii* Herbich var. *latilobum* (Maxim.) Kitamura (3:1) 25 mg/kg b.w.; ASC50, Ashwagandha (*Withania somnifera*): *Chrysanthemum zawadskii* Herbich var. *latilobum* (Maxim.) Kitamura (3:1) 50 mg/kg b.w.; ASC100, Ashwagandha (*Withania somnifera*): *Chrysanthemum zawadskii* Herbich var. *latilobum* (Maxim.) Kitamura (3:1) 100 mg/kg b.w. Values are presented as mean ± standard deviation (*n* = 8), and different superscript letters indicate significance at *p* < 0.05.

**Table 2 ijms-26-05230-t002:** Effects of ASC complex on biochemical factors of serum, feces, and white adipose tissue in HFD-induced obese mice.

	NC	CON	MF	AS25	C25	ASC25	ASC50	ASC100
Serum								
ALT (mU/mL)	51.04 ± 5.98 ^c^	99.90 ± 11.16 ^a^	62.79 ± 9.37 ^bc^	75.85 ± 15.25 ^b^	77.84 ± 12.01 ^b^	69.63 ± 8.38 ^b^	61.98 ± 8.14 ^bc^	63.36 ± 10.60 ^bc^
AST (mU/mL)	94.15 ± 5.97 ^c^	138.38 ± 5.63 ^a^	113.66 ± 9.38 ^b^	113.15 ± 13.65 ^b^	120.13 ± 8.45 ^b^	109.71 ± 4.55 ^b^	110.97 ± 11.58 ^b^	109.02 ± 7.15 ^b^
Triglyceride (mM)	4.83 ± 0.77 ^e^	26.34 ± 2.06 ^a^	8.92 ± 0.65 ^e^	17.03 ± 3.97 ^c^	20.87 ± 1.29 ^b^	16.78 ± 2.52 ^c^	13.68 ± 2.16 ^d^	11.88 ± 0.76 ^de^
Total chol (μg/μL)	9.31 ± 6.60 ^e^	46.90 ± 5.77 ^a^	24.83 ± 5.63 ^d^	41.38 ± 6.17 ^ab^	43.22 ± 9.42 ^ab^	37.24 ± 7.55 ^bc^	36.32 ± 5.63 ^bcd^	28.97 ± 3.78 ^cd^
LDL-chol (μg/μL)	0.23 ± 0.01 ^e^	0.76 ± 0.08 ^a^	0.32 ± 0.05 ^d^	0.51 ± 0.07 ^b^	0.57 ± 0.07 ^b^	0.50 ± 0.05 ^bc^	0.49 ± 0.06 ^bc^	0.41 ± 0.01 ^c^
HDL-chol (μg/μL)	1.42 ± 0.07 ^c^	2.45 ± 0.19 ^a^	1.87 ± 0.10 ^b^	1.97 ± 0.12 ^b^	2.06 ± 0.02 ^b^	2.08 ± 0.13 ^b^	1.97 ± 0.36 ^b^	1.86 ± 0.18 ^b^
LDL/HDL ratio	0.16 ± 0.01 ^d^	0.31 ± 0.03 ^a^	0.17 ± 0.03 ^d^	0.26 ± 0.04 ^bc^	0.28 ± 0.03 ^ab^	0.24 ± 0.02 ^bc^	0.25 ± 0.03 ^bc^	0.22 ± 0.01 ^c^
FFA (mM)	0.29 ± 0.03 ^d^	0.59 ± 0.06 ^a^	0.35 ± 0.04 ^cd^	0.44 ± 0.04 ^b^	0.52 ± 0.06 ^a^	0.44 ± 0.07 ^b^	0.44 ± 0.08 ^b^	0.41 ± 0.05 ^bc^
Glucose (nmol/μL)	11.68 ± 0.86 ^g^	21.28 ± 1.51 ^a^	13.13 ± 0.87 ^f^	18.50 ± 0.85 ^bc^	19.75 ± 1.04 ^a^	17.81 ± 1.01 ^c^	16.13 ± 0.59 ^d^	14.85 ± 0.57 ^e^
Insulin (ng/mL)	0.25 ± 0.07 ^g^	0.91 ± 0.07 ^a^	0.40 ± 0.07 ^f^	0.65 ± 0.02 ^c^	0.75 ± 0.06 ^b^	0.60 ± 0.01 ^cd^	0.54 ± 0.02 ^de^	0.51 ± 0.02 ^e^
HbA1c (%)	4.57 ± 0.26 ^d^	11.14 ± 0.68 ^a^	5.58 ± 0.57 ^e^	8.51 ± 0.49 ^c^	10.03 ± 0.65 ^b^	8.38 ± 0.25 ^c^	7.50 ± 0.76 ^cd^	7.03 ± 0.66 ^d^
Feces								
Triglyceride (mM)	1.34 ± 0.33 ^f^	2.17 ± 0.78 ^e^	4.97 ± 0.41 ^b^	3.37 ± 0.46 ^cd^	3.07 ± 0.33 ^d^	3.77 ± 0.39 ^c^	5.14 ± 0.47 ^b^	6.44 ± 0.42 ^a^
Total chol (μg/mL)	0.39 ± 0.19 ^e^	0.59 ± 0.22 ^d^	1.36 ± 0.12 ^b^	1.00 ± 0.16 ^c^	0.91 ± 0.19 ^c^	1.05 ± 0.05 ^c^	1.39 ± 0.19 ^b^	1.70 ± 0.14 ^a^

ALT, alanine aminotransferase; AST, aspartate aminotransferase; Total chol, total cholestrol; LDL-chol, low-density lipoprotein-cholesterol; HDL-chol, high-density lipoprotein-cholesterol; FFA, free fatty acid; HbA1c, hemoglobin A1C. NC, normal control; CON, obesity-induced control; MF, Metformin 250 mg/kg b.w., positive control; AS25, Ashwagandha (*Withania somnifera*) 25 mg/kg b.w., C25, *Chrysanthemum zawadskii* Herbich var. *latilobum* (Maxim.) Kitamura 25 mg/kg b.w.; ASC25, Ashwagandha (*Withania somnifera*): *Chrysanthemum zawadskii* Herbich var. *latilobum* (Maxim.) Kitamura (3:1) 25 mg/kg b.w.; ASC50, Ashwagandha (*Withania somnifera*): *Chrysanthemum zawadskii* Herbich var. *latilobum* (Maxim.) Kitamura (3:1) 50 mg/kg b.w.; ASC100, Ashwagandha (*Withania somnifera*): *Chrysanthemum zawadskii* Herbich var. *latilobum* (Maxim.) Kitamura (3:1) 100 mg/kg b.w. Values are presented as mean ± standard deviation (*n* = 8), and different superscript letters indicate significance at *p* < 0.05.

## Data Availability

The data presented in this study are available on request from the corresponding author.

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
