# Peer review of "Withania somnifera and Chrysanthemum zawadskii Herbich var. latilobum (Maxim.) Kitamura Complex Attenuates Obesity in High-Fat-Diet-Induced Obese Mice"

_ijms, 2025, doi:10.3390/ijms26115230_

Round 1
Reviewer 1 Report
Comments and Suggestions for Authors
Overall, an excellent manuscript is submitted. Exploration of the effects of extracts from medicinal plants in preclinical pharmacological studies is very important as further development of natural products requires clear beneficial biological activity. The study design is appropriate. All conclusions done are supported by results. Figures and tables are informative and in good quality. English is acceptable. There are minor technical errors regarding botanical names of the plants (given in uploaded pdf). The manuscript could be accepted in its present form, after minor revisions of technical errors.

Author Response
Thank you very much for your thorough and thoughtful review of our manuscript. We truly appreciate the time and effort you dedicated to evaluating our work.
We have carefully reviewed all the comments and suggestions provided in the annotated PDF file. Based on your detailed feedback, we have revised the manuscript accordingly. Please find below a point-by-point response to each of the comments, along with a summary of the changes made.
We believe these revisions have significantly improved the clarity and quality of the manuscript. Should there be any further questions or recommendations, we would be happy to address them.
Thank you once again for your kind and meticulous review.
Reviewer 2 Report
Comments and Suggestions for Authors
The authors mention that Ashwagandha possesses anti-inflammatory and antioxidant properties. However, it would be helpful if the authors could elaborate on the specific signaling pathways through which these effects are mediated, to provide a clearer mechanistic understanding.
The authors have conducted extensive western blot analyses to assess protein expression levels. It would be valuable to clarify why protein-level analysis was prioritized over gene-level analysis, given that gene expression changes typically precede alterations at the protein level.
Additionally, the MAPK signaling pathway plays crucial roles in various biological processes, including neural differentiation. Regarding the statement that “MAPK signaling influences adipogenesis by modulating the activity of PPAR-γ and C/EBP, thereby regulating extracellular signals into adipogenic differentiation,” it is recommended that the authors provide appropriate references to strengthen the credibility of this claim.
Author Response
We sincerely appreciate your careful and considerate review of our manuscript. Your insightful comments and suggestions were extremely helpful in guiding our revisions.
In response to your feedback, we have reviewed the manuscript thoroughly and compiled the suggested changes. We are pleased to submit our revised version accordingly.
Please find enclosed a detailed response addressing each of the comments. We have done our best to reflect your recommendations and improve the overall quality and clarity of the manuscript.
Thank you once again for your time and valuable input.
- The authors mention that Ashwagandha possesses anti-inflammatory and antioxidant properties. However, it would be helpful if the authors could elaborate on the specific signaling pathways through which these effects are mediated, to provide a clearer mechanistic understanding.
- We appreciate the reviewer’s thoughtful suggestion. However, we respectfully clarify that the primary objective of the present study was not to evaluate the anti-inflammatory or antioxidant effects of Ashwagandha. Rather, our experimental design focused specifically on investigating its anti-obesity potential in high-fat diet-induced obese mice. Accordingly, the results and discussion were structured around the mechanisms involved in adipogenesis, lipid metabolism, lipolysis, and energy expenditure, which are more directly relevant to obesity-related pathways. That said, to enhance the scientific context and provide a more comprehensive understanding, we have added a brief explanation of the signaling pathways known to mediate Ashwagandha’s anti-inflammatory and antioxidant effects (e.g., NF-κB, Nrf2, MAPK) in the revised Introduction section. We hope this clarification adequately addresses the reviewer’s concern.
- The authors have conducted extensive western blot analyses to assess protein expression levels. It would be valuable to clarify why protein-level analysis was prioritized over gene-level analysis, given that gene expression changes typically precede alterations at the protein level.
- We appreciate this insightful point. Protein-level analysis was prioritized in our study because proteins are the direct functional molecules involved in metabolic regulation. More importantly, post-translational modifications such as phosphorylation (e.g., p-AMPK, p-ACC, p-CREB) are critical for functional activity and cannot be captured by mRNA expression alone. By analyzing protein levels, particularly their active (phosphorylated) forms, we were able to assess biological activity more directly.
- Additionally, the MAPK signaling pathway plays crucial roles in various biological processes, including neural differentiation. Regarding the statement that “MAPK signaling influences adipogenesis by modulating the activity of PPAR-γ and C/EBP, thereby regulating extracellular signals into adipogenic differentiation,” it is recommended that the authors provide appropriate references to strengthen the credibility of this claim.
- We thank the reviewer for this important suggestion. To support our statement, we have now cited relevant studies demonstrating that MAPK signaling modulates adipogenesis through regulation of transcription factors PPAR-γ and C/EBPα. Specifically, ERK1/2 activation has been shown to influence early adipogenic commitment and differentiation (Lefterova et al., Trends Endocrinol Metab, 2014)